# When LLM Meets Time Series: Can LLMs Perform Multi-Step Time Series Reasoning and Inference

## Abstract

The rapid advancement of Large Language Models (LLMs) has sparked growing interest in their application to time series analysis. Yet, their ability to perform complex reasoning over temporal data remains underexplored. A rigorous benchmark is a crucial first step toward systematic evaluation. In this work, we present the **TSAIA Benchmark**, a comprehensive framework for assessing LLMs as time-series artificial intelligence assistants. TSAIA integrates two complementary tiers of tasks. The *series-centric* tier instantiates canonical time-series formulations—such as forecasting, anomaly detection, and risk-return analysis—via a controlled question-generation pipeline, providing continuity with prior evaluation settings. The *problem-centric* tier, in contrast, derives tasks from real-world analytical questions in healthcare, retail, and climate science, and formalizes their construction through a task-design paradigm spanning three levels: *evidence integration*, *operator-based comparison*, and *structural multi-step reasoning*. This paradigm enables dynamic extensibility, allowing new task instances to be generated as data evolve in practice. To accommodate heterogeneous task types, we define task-specific success criteria and tailored inference quality metrics, applied under a unified evaluation protocol. We evaluate 7 state-of-the-art LLMs and find that while they achieve reasonable performance on series-centric tasks, they struggle substantially on problem-centric ones, often failing at multi-step reasoning, numerical precision, and constraint adherence. These results underscore the need for domain-grounded, dynamically extensible benchmarks as a foundation for advancing LLM-based time-series assistants.

## 1 Introduction

Large Language Models (LLMs) have demonstrated remarkable general-purpose capabilities across various domains, such as language understanding Dong et al. (2019), code generation Jiang et al. (2024), and scientific reasoning Taylor et al. (2022); Wang et al. (2025c). However, their ability to perform complex reasoning over *time series data* remains significantly underexplored. Time series analysis is a fundamental competency for data analysts and scientists in fields like energy Alvarez et al. (2010), finance Sezer et al. (2020), climate Mudelsee (2019), and healthcare Rathlev et al. (2007), yet it remains an area where LLMs are relatively untested. In practice, real-world time series workflows are inherently complex Yan et al. (2021); Han et al. (2021): they require **multi-step reasoning** Fu et al. (2022), precise numerical computation Cvejoski et al. (2022), integration of domain knowledge Xue et al. (2024), and adherence to operational constraints Wang et al. (2016). With the rise of LLM-based agents Li et al. (2025); Chang et al. (2025), there is growing interest in developing intelligent systems that can interpret natural language instructions for time series analysis. However, since time series analysis is challenging even for humans Uddin et al. (2024), a rigorous benchmark is necessary to evaluate whether LLMs can truly serve as reliable time series reasoning assistants.

Existing benchmarks have made progress toward temporal or time-series evaluation, but they remain insufficient. Many focus on individual subtasks Merrill et al. (2024) or fixed experimental configurations (e.g., sliding-window forecasting Du et al. (2024)), or reduce the problem to simple question-answering (QA) tasks that use temporal keywords, but do not require structured numerical

reasoning Zhou et al. (2021). Others fail to incorporate realistic operational constraints Abdullahi et al. (2025) or to capture the inherently multi-step nature of time series workflows Chang et al. (2025). Critically, most existing datasets are static Du et al. (2024); Liu et al. (2024b); Xu et al. (2024), limiting their ability to reflect evolving real-world settings and preventing long-term evaluation of models' adaptivity.

To address these limitations, we present the **Time Series Artificial Intelligence Assistant (TSAIA )** benchmark for evaluating LLMs as general-purpose time-series assistants. TSAIA is grounded in real-world datasets across healthcare, retail, climate, energy, and finance , and it combines two complementary suites of tasks. The *series-centric* suite instantiates canonical time-series formulations via template-defined constructions (e.g., forecasting, anomaly detection, analytical and decision-oriented tasks), providing a controlled baseline and continuity with prior evaluation settings . The *problem-centric* suite is derived from researcher-driven questions in practical domains and is instantiated through a task-design paradigm that supports continual creation of task instances as new data arrive . This paradigm structures multi-step reasoning along three complementary levels: *Evidence*—integration of heterogeneous variables or modalities before inference; *Operator*—after temporal understanding, application of mathematically specified computations or comparisons over features (across series or within a series); and *Structural*—execution of ordered, compositional pipelines with intermediate artifacts (e.g., detection → attribution → forecasting). A unified evaluation protocol applies to both suites, enabling baseline assessment on series-centric tasks and rigorous, domain-grounded appraisal of multi-step reasoning on problem-centric tasks. To achieve good performance on TSAIA, the following capabilities are needed: compositional reasoning Li et al. (2024) (the sequential execution of logical and numerical operations to construct end-to-end analytical pipelines), comparative reasoning (selecting the optimal asset based on calculated summary indicators), commonsense reasoning Davis & Marcus (2015) (identifying plausible covariates for the target variable), decision-oriented reasoning (interpreting risk-return metrics in investment contexts), and numerical precision.

We evaluate a set of state-of-the-art LLMs under this benchmark, including GPT-4o Hurst et al. (2024), Qwen-Max Qwen Team (2025), Claude-3.7 Sonnet Anthropic (2025), Gemini-2.5 Google DeepMind (2025),Grok-4 xAI (2025) and others, using a unified agent framework Gao et al. (2024) that prompts models to generate executable code and iteratively refine their predictions. Our findings show that while LLMs can succeed on core static tasks, they struggle significantly on dynamic, real-world multi-step tasks, often failing to integrate constraints, carry out intermediate reasoning, or adapt under distribution shifts Fan et al. (2023). These results underscore the urgent need for domain-grounded, dynamically extensible benchmarks, and position TSAIA as a foundation for the development of reasoning-capable time series AI assistants.

## 2 RELATED WORK

| Benchmark | Dynamic | TS-Involved | Reasoning | #Tasks | Task Type |
|---|---|---|---|---|---|
| Test of Time Fatemi et al. (2024) | ✗ | ✗ | ✓ | 1 | QA |
| TRAM Wang & Zhao (2024) | ✗ | ✗ | ✓ | 1 | QA |
| TSI-Bench Du et al. (2024) | ✗ | ✓ | ✗ | 1 | TS Analysis |
| TSB-AD Liu & Paparrizos (2024) | ✗ | ✓ | ✗ | 1 | TS Analysis |
| GIFT-Eval Aksu et al. (2024) | ✗ | ✓ | ✗ | 1 | TS Analysis |
| TFB Qiu et al. (2024) | ✗ | ✓ | ✗ | 1 | TS Analysis |
| Time-MMD Liu et al. (2024b) | ✗ | ✓ | ✗ | 1 | TS Analysis |
| CiK Williams et al. (2024) | ✗ | ✓ | ✗ | 1 | TS Analysis |
| TGTSF Xu et al. (2024) | ✗ | ✓ | ✗ | 1 | TS Analysis |
| LLM TS Struggle Merrill et al. (2024) | ✗ | ✓ | ✓ | 2 | QA, TS Analysis |
| MTBench Chen et al. (2025) | ✗ | ✓ | ✓ | 3 | QA, TS Analysis |
| ChatTime Wang et al. (2025a) | ✗ | ✓ | ✓ | 3 | QA, TS Analysis |
| **TSAIA(Ours)** | ✓ | ✓ | ✓ | 4+, extensible | QA, TS Analysis |

Table 1: Comparison of TSAIA and existing temporal-related benchmarks. Dynamic indicates whether new task instances can be continuously generated.

## 2.1 BENCHMARKS FOR TEMPORAL AND TIME-SERIES REASONING

A number of recent efforts examine LLMs' capabilities on temporal or time-series related tasks, yet they differ markedly from the goal of evaluating multi-step, constraint-aware *time-series assistants* in real-world workflows. Test of Time (ToT) Fatemi et al. (2024) focuses on temporal reasoning with synthetic datasets that control factors such as fact order, graph structure, and temporal arithmetic. TRAM Wang & Zhao (2024) aggregates ten textual temporal-reasoning datasets spanning order, duration, frequency, and arithmetic, reporting substantial gaps to human performance. Closer to classical time-series pipelines, TSI-Bench Du et al. (2024) targets imputation across real datasets and models with standardized evaluation, while TSB-AD Liu & Paparrizos (2024) curates a large, heterogeneous anomaly-detection suite and argues for more reliable metrics such as VUS-PR. These benchmarks either center on textual temporal logic or isolate single TS subtasks (e.g., imputation, anomaly detection), and typically use static datasets and single-step or end-only evaluation protocols. By contrast, TSAIA is introduced as a *domain-grounded benchmark* with two complementary tiers: a *series-centric Tier* that instantiates canonical time-series tasks for continuity and baseline comparison, and a *problem-centric Tier* that derives real-world analytical tasks across domains, formalized through operator-based design and intermediate-step evaluation, enabling dynamic extensibility as data evolve.

## 2.2 MULTI-STEP BENCHMARKS IN OTHER DOMAINS: A FOUR-LEVEL TAXONOMY

Beyond time series, multi-step reasoning has been extensively explored under diverse task constructions. We summarize prominent *task construction paradigms* into four levels and draw from the first three to define TSAIA's dataset-generation paradigm.

**Evidence Level — Cross-document / Cross-modal Aggregation (Multi-variant tasks).** Tasks require aggregating evidence across multiple documents or modalities. Representative datasets include HotpotQA for multi-hop QA with supporting-fact supervision Yang et al. (2018)and ScienceQA for multimodal science questions with lectures and explanations Saikh et al. (2022). These works emphasize evidence collection/aggregation and often supervise supporting facts or interaction traces, inspiring our multi-parameter TS tasks that aggregate signals across series, covariates, and modalities.

**Operator Level — Numerical / Symbolic Computation after Understanding (Operation tasks).** Tasks explicitly require discrete operations or comparisons after comprehension. Canonical datasets include DROP for discrete reasoning over paragraphs Dua et al. (2019) and TAT-QA for table-and-text financial QA with numerical operations Zhu et al. (2021). This line motivates TSAIA's operator-level design, where time-series problems require computing indicators or comparing outcomes, such as evaluating which patient faces higher mortality risk, or contrasting climate anomalies across regions.

**Structural Level — Constraining and Supervising Intermediate Steps (Multi-step tasks).** Here, task definitions scaffold or supervise the *sequence* of reasoning steps. QDMR/BREAK introduces a decomposition meaning representation with step-annotated questions Wolfson et al. (2020), while ProofWriter evaluates logical chains with structured intermediate proofs Tafjord et al. (2020). TSAIA adapts this idea to time series by formalizing structural stages (e.g., detection $\rightarrow$ attribution $\rightarrow$ forecasting) with intermediate outputs and explicit checks.

**Process Level — Action Sequences (Agent-style loops).** Some benchmarks evaluate agents that perform multi-step plans in interactive environments, such as GAIA Mialon et al. (2023) and AgentBench Liu et al. (2023). While our focus is not on full agent loops, these works highlight the broader direction of process-oriented evaluation, where models must plan, execute, and adapt their reasoning pipelines over evolving contexts.

# 3 BENCHMARK PARADIGM

## 3.1 DATA SOURCES

To evaluate time series AI assistants effectively, we focus on tasks grounded in real-world use cases that data analysts across diverse application domains routinely encounter. Our objective is to assess whether an assistant can handle practical, multi-step scenarios involving time series, thereby ensuring its utility in everyday analytical workflows. We reviewed over twenty research publications addressing time-series challenges in healthcare, finance, energy, climate science, and

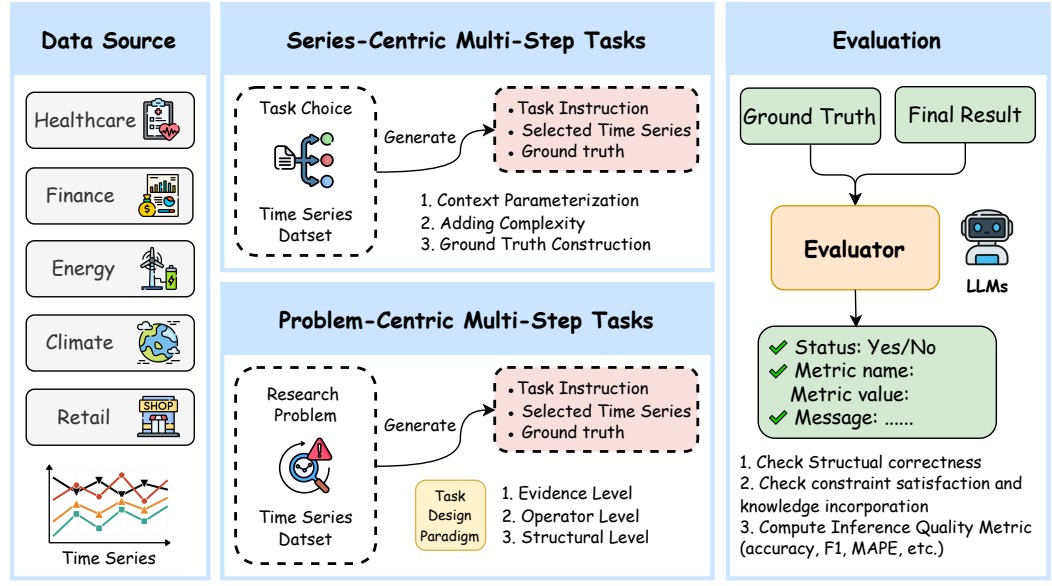

Figure 1: Pipeline for multi-step time-series task construction and instance generation, followed by unified evaluation.

retail, and extracted recurring problems that involve reasoning and multi-stage analytical pipelines, converting them into benchmark tasks. To ensure breadth and domain grounding, we curated datasets from a wide range of sources, including the MIMIC-IV databaseJohnson et al. (2020) with detailed ICU records such as vital signs, laboratory results, and outcomes; the FreshRetailNet-50K dataset Wang et al. (2025b)capturing product-level demand, pricing, discounts, and holiday effects across retail settings; and climate resources such as the ERA5 reanalysisHersbach et al. (2018) and MTBenchChen et al. (2025), which provide multimodal meteorological variables including temperature, precipitation, and wind. We also incorporated energy datasets such as electricity load, renewable generation, and building consumption, as well as financial market data spanning indices and individual equities, alongside physiological recordings such as ECG from public repositories. All sources are preprocessed into aligned input–output windows and enriched with standardized metadata covering units, covariates, temporal granularity, and operational context, ensuring that task instances are consistent and comparable across domains. This heterogeneous but coherent collection provides the foundation for constructing benchmark tasks that reflect both the intrinsic statistical properties of time series and the domain-specific reasoning challenges faced by practitioners.

## 3.2 SERIES-CENTRIC MULTI-STEP TASKS

This component focuses on series-centric formulations that primarily probe the statistical and structural properties of time series themselves. Tasks are organized along four families reflecting common analytical workflows: *predictive* (e.g., constraint-aware forecasting under operational limits), *diagnostic* (e.g., anomaly detection and causal attribution with priors), *analytical* (e.g., portfolio risk–return analytics, backtesting of rule-based strategies), and *decision-making* (e.g., selecting among structured indicators under explicit criteria). These families abstract recurring practitioner questions while remaining faithful to canonical time-series operations.

**Task Construction.** Following the original design, each family is specified by natural-language templates that determine the observed window, forecast/decision horizon, admissible covariates, units, and (when applicable) operational constraints (e.g., non-negativity, ramp-rate, or clinical thresholds). Dataset slices are sampled under fixed time cuts to avoid leakage and to preserve reproducibility. Templates are then parameterized by the sampled context (time spans, variable names, covariate lists) and rendered into instructions that are unambiguous with respect to I/O schema.

**Instance Materialization & Ground Truth.** Instances consist of (i) an instruction, (ii) serialized inputs with metadata, and (iii) a verifiable reference output. Ground truth is bound deterministically:

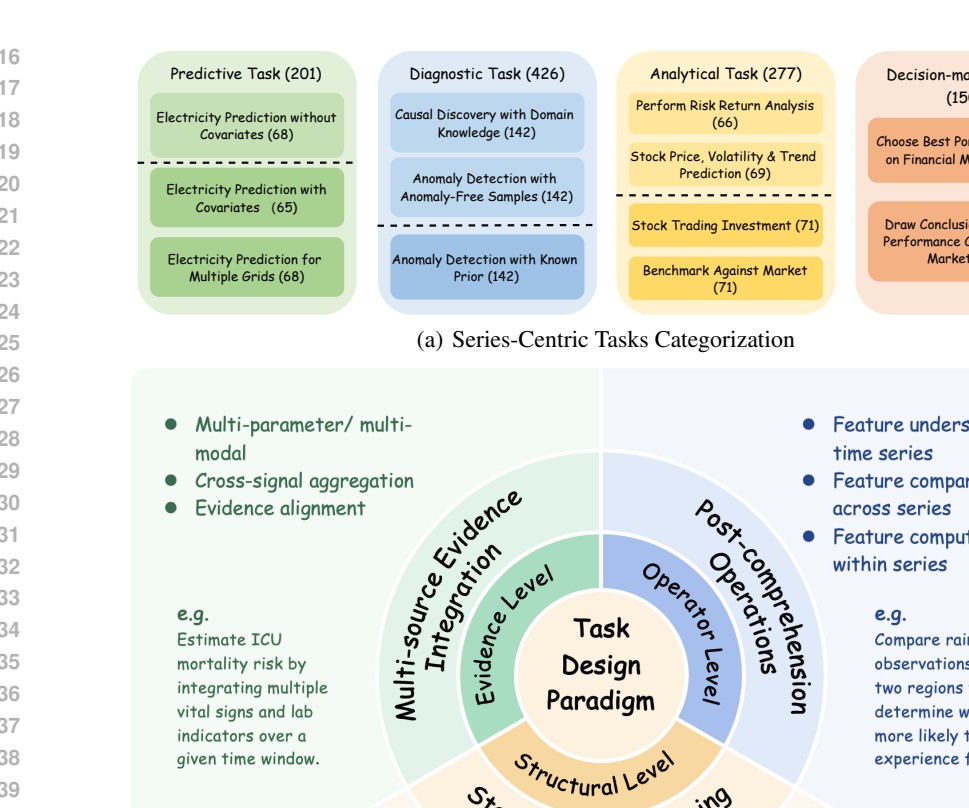

(a) Series-Centric Tasks Categorization

(b) Problem-Centric Tasks Design Paradigm

Figure 2: Illustrations of two task design. (a) Categorization of Series-Centric Tasks in TSAIA. Lighter colors denote tasks with less difficulty and darker colors denote tasks with higher difficulty. (b) Problem-Centric Task Design Paradigm across three levels: Evidence, Operator, and Structural.

either by retrieving targets from held-out future windows, or by applying reference computations (e.g., statistical indicators, portfolio functions) defined by the template. The resulting series-centric tier provides a controlled, reproducible baseline for assessing whether LLMs can execute well-posed time-series tasks with numerical precision and format correctness.

## 3.3 PROBLEM-CENTRIC MULTI-STEP TASKS

Complementing the above, this component derives tasks directly from questions that domain researchers genuinely care about in healthcare, retail, and climate settings. Task generation follows our task design paradigm across three complementary levels that emphasize multi-step reasoning beyond isolated statistical operations.

**Evidence Level.** Tasks require integrating heterogeneous variables/modalities and aligning them into a sufficient evidence set prior to inference (e.g., combining precipitation, temperature, and wind to identify extreme weather anomalies).

**Operator Level.** Tasks require feature understanding of time series followed by explicit operations on derived quantities, including feature comparison across series or feature computation within a series (e.g., deciding which ICU patient has higher near-term mortality risk from trajectories).

| Time | Tempe -rature | Relative Humidity | Wind Speed | Load Power |
|------|------|------|------|------|
| 2020-09-13 09:44 | 24.58 | 89.41 | 1.4 | 0.923 |
| 2020-09-13 09:45 | 24.60 | 89.31 | 1.4 | 0.924 |
| ... | ... | ... | ... | ... |
| 2020-09-13 11:39 | 25.40 | 81.87 | 1.4 | 1.003 |
| 2020-09-13 11:40 | 25.40 | 81.87 | 1.4 | 1.004 |

**Task Instruction**

I have historical Temperature, Relative Humidity, Wind Speed data and the corresponding load_power data for the past 117 minutes. I need to ensure that the maximum allowable system load does not exceed 1.0689227278350713 MW. Think about how Temperature, Relative Humidity, Wind Speed influence load_power. Please give me a forecast for the next 12 minutes for load_power. Your goal is to make the most accurate forecast as possible, refine prediction result based on the constraint previously described, and ...

**Ground Truth**

1.0051
1.0057
1.0062
1.0068
1.0073
1.0079
1.0084
1.0090
1.0095
1.0101
1.0106
1.0112

Figure 3: Example Task Instance containing the task instruction, accompanied serialized dataset, and ground truth.

**Structural Level.** Tasks require executing an ordered, compositional pipeline with intermediate artifacts and stepwise checks (e.g., demand recovery $\rightarrow$ substitution estimation $\rightarrow$ long-horizon forecasting).

The paradigm maps real questions into tasks by extracting relevant data subsets and encoding domain constraints into the instruction. Ground truth is established through later observations when available, or through deterministic procedures aligned with the template. When questions involve future outcomes, labels are revealed gradually, allowing longitudinal evaluation.

### 3.4 EVALUATION PROTOCOL

Evaluation in TSAIA must accommodate heterogeneous task goals while maintaining comparability. Each task type is paired with success criteria and inference-quality metrics aligned with practical expectations (e.g., MAE/RMSE/MAPE for forecasting, precision/recall-based statistics such as VUS-PR for anomaly detection, AUROC/accuracy for classification and decision-oriented tasks). Across all tasks, outputs must conform to the expected format, satisfy injected constraints, and appropriately incorporate provided domain knowledge. Trivial or degenerate responses—such as constant predictions, all-zero anomaly labels, or invalid code—are flagged as failures even when syntactically well-formed.

The evaluation proceeds in three stages: (i) validation of structural correctness and shape conformity; (ii) verification of constraint satisfaction and the use of required auxiliary information; and (iii) computation of task-specific metrics against ground truth. Results are returned in a structured record with overall success status, diagnostic messages, and detailed metric scores. Failures are categorized into execution errors (outputs cannot be parsed or executed), constraint violations (outputs contradict injected rules), and low-quality predictions (outputs meet format requirements but fall below metric thresholds). For problem-centric tasks whose labels are revealed over time, we additionally track performance longitudinally to assess adaptivity under distributional shifts.

## 4 EXPERIMENTS

### 4.1 EXPERIMENTAL SETUP

We evaluate the performance of seven large language models (LLMs) on the full suite of benchmark tasks: GPT-4o Hurst et al. (2024), Qwen-Max Qwen Team (2025), Claude-3.7 Sonnet Anthropic (2025), DeepSeek Liu et al. (2024a), Gemini-2.5 Google DeepMind (2025), Codestral AI (2024), and Grok-4 xAI (2025). Among them, Codestral is a code-specialized model built upon Mistral Mistral AI (2024). To address LLMs' limitations in processing structured numerical inputs and

| Task | Metric | GPT-4o | Qwen-Max | Claude-3.7 | DeepSeek | Gemini-2.5 | Codestral | Grok-4 |
|------|--------|--------|----------|-----------|----------|-----------|-----------|--------|
| **Predictive Tasks** | | | | | | | | |
| Elec. Pred. w/ Cov. (Min Load) | Success Rate | 0.76 | 0.82 | 0.64 | 0.88 | 0.70 | 0.59 | 0.88 |
| | MAPE (std) | 0.11 (0.11) | 0.09 (0.11) | 0.07 (0.10) | 0.12 (0.18) | 0.09 (0.11) | 0.09 (0.10) | 0.06 (0.09) |
| Elec. Pred. w/o Cov. (Load Var.) | Success Rate | 0.82 | 0.88 | 0.06 | 0.76 | 0.11 | 0.71 | 1.00 |
| | MAPE (std) | 0.17 (0.12) | 0.13 (0.09) | 0.34 (0.0) | 0.19 (0.17) | 0.20 (0.14) | 0.13 (0.07) | 0.08 (0.10) |
| **Diagnostic Tasks** | | | | | | | | |
| Extreme Weather Detection | Success Rate | 0.24 | 0.23 | 0.51 | 0.23 | 0.42 | 0.23 | 0.34 |
| | F1 (std) | 0.91 (0.23) | 0.90 (0.24) | 0.90 (0.20) | 0.90 (0.24) | 0.73 (0.25) | 0.91 (0.23) | 0.91 (0.20) |
| ECG Anomaly | Success Rate | 0.51 | 0.17 | 0.58 | 0.54 | 0.35 | 0.59 | 0.76 |
| | F1 (std) | 0.55 (0.35) | 0.70 (0.29) | 0.67 (0.25) | 0.54 (0.34) | 0.76 (0.21) | 0.58 (0.34) | 0.80 (0.16) |
| Energy Usage Anomaly | Success Rate | 0.87 | 0.52 | 0.79 | 1.00 | 0.50 | 0.58 | 0.97 |
| | F1 (std) | 0.08 (0.09) | 0.14 (0.20) | 0.49 (0.18) | 0.50 (0.19) | 0.33 (0.22) | 0.06 (0.06) | 0.42 (0.21) |
| **Analytical Tasks** | | | | | | | | |
| Future Price | Success Rate | 0.96 | 1.00 | 0.70 | 0.87 | 0.70 | 0.39 | 1.00 |
| | MAPE (std) | 0.06 (0.08) | 0.05 (0.07) | 0.21 (0.38) | 0.05 (0.07) | 0.03 (0.03) | 0.05 (0.05) | 0.06 (0.07) |
| Sharpe Ratio | Success Rate | 0.73 | 0.18 | 0.09 | 0.18 | 0.09 | 0.73 | 0.09 |
| | Abs Error (std) | 0.00 (0.00) | 0.00 (0.00) | 0.03 (0.0) | 0.02 (0.02) | 0.03 (0.0) | 0.00 (0.00) | 0.03 (0.0) |

Table 2: Selected results on Series-Centric Tasks. "Elec. Pred. w/ Cov." = Electricity Prediction with Covariates; "Elec. Pred. w/o Cov." = Electricity Prediction without Covariates; We report representative tasks from each family while preserving all metrics. Full results are in the Appendix.**??**

| Task | Metric | GPT-4o | Qwen-Max | Claude-3.7 | DeepSeek | Gemini-2.5 | Codestral | Grok-4 |
|------|--------|--------|----------|-----------|----------|-----------|-----------|--------|
| **Retail Tasks** | | | | | | | | |
| Products Substitution-Effect Ranking | Success Rate | 0.64 | 0.59 | 0.98 | 1.00 | 0.34 | 0.67 | 0.96 |
| | NDCG@3 (std) | 0.74 (0.26) | 0.74 (0.25) | 0.72 (0.07) | 0.70 (0.29) | 0.47 (0.45) | 0.71 (0.25) | 0.69 (0.28) |
| | MAP@3 (std) | 0.82 (0.24) | 0.83 (0.24) | 0.83 (0.05) | 0.82 (0.23) | 0.78 (0.26) | 0.81 (0.25) | 0.79 (0.28) |
| Customer Demand Forecast | Success Rate | 0.76 | 0.82 | 0.58 | 0.86 | 0.73 | 0.88 | 0.88 |
| | WPE (std) | 0.14 (1.51) | 0.22 (1.69) | 0.69 (2.35) | 0.80 (2.31) | 0.70 (2.31) | 0.08 (1.40) | 0.68 (2.15) |
| | WAPE (std) | 1.54 (1.07) | 1.61 (1.22) | 1.98 (1.85) | 2.00 (1.81) | 2.06 (1.91) | 1.51 (0.96) | 1.93 (1.72) |
| **Healthcare Tasks** | | | | | | | | |
| ICU Patient Death Risk Evaluation | Success Rate | 0.96 | 0.94 | 0.82 | 1.00 | 0.82 | 0.96 | 1.00 |
| | MSE (std) | 0.86 (0.86) | 1.74 (5.62) | 0.63 (0.74) | 0.84 (0.81) | 0.73 (0.73) | 0.74 (0.80) | 0.90 (0.81) |
| Healthcare Task One | Success Rate | 1.00 | 0.98 | 1.00 | - | 0.98 | 0.98 | 1.00 |
| | Acc (std) | 0.56(0.49) | 0.45 (0.50) | 0.44 (0.50) | - | 0.44 (0.50) | 0.53(0.49) | 0.62 (0.49) |
| **Climate Tasks** | | | | | | | | |
| Weather Forecasting | Success Rate | 0.94 | 1.00 | 0.90 | 1.00 | 0.88 | 1.00 | 1.00 |
| | MAPE (std) | 0.25 (0.32) | 0.20 (0.24) | 0.13 (0.17) | 0.07 (0.03) | 0.26 (0.28) | 0.29 (0.51) | 0.09 (0.20) |
| Property Damage Prediction | Success Rate | 1.00 | 1.00 | 0.84 | 1.00 | 0.92 | 1.00 | 1.00 |
| | Accuracy (std) | 0.39 (0.36) | 0.49 (0.36) | 0.34 (0.31) | 0.34 (0.29) | 0.26 (0.30) | 0.35 (0.35) | 0.35 (0.35) |
| Social Impact Prediction | Success Rate | 1.00 | 1.00 | 0.82 | 0.98 | 0.88 | 1.00 | 1.00 |
| | Accuracy (std) | 0.20 (0.28) | 0.36 (0.40) | 0.42 (0.37) | 0.40 (0.35) | 0.51 (0.31) | 0.29 (0.36) | 0.42 (0.30) |

Table 3: Model Performance on **Problem-Centric Tasks**, covering retail, healthcare, and climate domains. The **Metric** column specifies the evaluation measure used for each subtask. Red indicates the best result, and Blue indicates the second best among populated entries.

producing high-precision, correctly shaped numerical outputs, we adopt the CodeAct Wang et al. (2024) agent framework for all models. Agentscope CodeAct agent Gao et al. (2024) enables code-based interaction by allowing LLMs to generate executable Python code, receive execution feedback, and revise outputs accordingly. All models are accessed via their official APIs and ran on a single NVIDIA A40 GPU with 48G memory. For all experiments, we use the same hyperparameters, temperature=0.0 for the most deterministic output and top_p=1.0.

For each benchmark task, models are provided with the same input data, including a task instruction and a serialized time series dataset in .pkl format. Responses are executed within a controlled jupyter notebook python interpreter provided by CodeAct agent setup. The final outputs are passed to task-specific evaluators, which extract predictions and compute metrics based on ground truth labels or evaluation programs. Each model's performance is assessed using two main criteria: The primary metric is Success Rate which is defined as the proportion of task instances for which the model output satisfies the predefined success criteria (see Table 4). For outputs deemed successful, we further evaluate quality using task-specific metrics (e.g., MAPE for forecasting, F1-score for anomaly detection), providing a more fine-grained comparison of inference quality.

| Task Type | Success Criterion | Metrics |
|---|---|---|
| Constrained Forecasting | Prediction is of correct shape and satisfies the specified operational constraint and the prediction is non-trivial (MAPE<1) | MAPE, WPE, WAPE |
| Anomaly Detection w/ reference samples | A binary sequence with correct length is obtained and the prediction is non-trivial (F1-score>0) | F1-score |
| Causal Inference w/ domain knowledge | A binary causal matrix with correct shape is returned. The provided domain knowledge is incorporated | Accuracy |
| Financial Analytics | A scalar value is returned and the prediction is non-trivial (absolute error<0.05) | Absolute Error |
| Financial Trading | An investment signal of correct length is returned and there is no loss in investment | CR, AR, MDD |
| Retail Product Substitution Ranking | A valid ranking list with all candidate products included exactly once, in correct format, and non-degenerate scores | NDCG, MAP |

Table 4: Task-specific success criteria and inference quality metrics. CR denotes Cumulative Return, AR denotes Annualized Return, MDD denotes Maximum Drawdown.

## 4.2 RESULT

We report results on both series-centric and problem-centric tasks. Tables 2 and 3 present the full quantitative outcomes.

**Series-Centric Tasks.** On canonical series-centric tasks such as electricity load forecasting and financial metric estimation, models generally achieve strong performance under simple formulations (e.g., maximum or minimum load). Success rates in these cases often exceed 0.9, although more complex variants—including ramp-rate control and multi-grid forecasting—remain challenging, with noticeably lower success rates and higher error measures. Similarly, diagnostic tasks (e.g., ECG anomaly detection, extreme weather detection) and analytical tasks (e.g., risk-return estimation, backtesting) expose significant variance in model capabilities.

**Problem-Centric Tasks.** In contrast, problem-centric tasks drawn from healthcare, retail, and climate domains highlight the limitations of LLMs when confronted with domain-grounded, multi-step reasoning. Retail subtasks such as product substitution ranking and latent demand forecasting reveal difficulties in compositional reasoning. In climate-related forecasting and impact estimation, success rates are relatively high, yet fine-grained accuracy remains inconsistent. Healthcare tasks are particularly challenging, as models often struggle to integrate heterogeneous ICU signals. Importantly, despite their higher intrinsic difficulty, problem-centric tasks achieve higher construction success rates, reflecting their grounding in well-defined domain problems that reduce ambiguity and enhance interpretability.

**Analysis** As summarized in Table 9, the proposed Problem-Centric Multi-Step Task Design introduces tasks that are derived directly from domain-grounded analytical questions rather than abstract time series manipulations. Figure 4.2 further contrasts the series-centric and problem-centric paradigms across different datasets and representative task types. We observe that, despite being

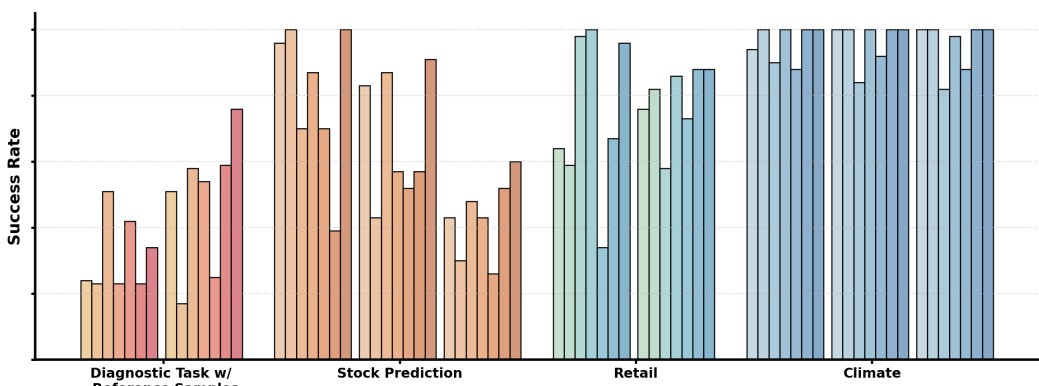

Figure 4: Visualization of success rates across four representative tasks. From left to right, the tasks are: *Diagnostic Task w/ Reference Samples*, *Stock Prediction*, *ReTail*, and *Climate*. For each task, bars are grouped by model in the order from left to right: GPT-4o, Qwen-Max, Claude-3.7, DeepSeek, Gemini-2.5, Codestral, and Grok-4. Warm low-saturation colors denote *Series-Centric* tasks (*Diagnostic Task w/ Reference Samples* and *Stock Prediction*), while cool low-saturation colors denote *Problem-Centric* tasks (*Retail* and *Climate*).

more challenging, the problem-centric paradigm achieves a higher overall task generation success rate.

Compared with the series-centric generation approach, our problem-centric paradigm produces tasks that are inherently more demanding due to their grounding in real-world analytical questions and multi-step reasoning requirements. Nevertheless, the generation process yields a higher success rate, since tasks are formulated from well-defined domain problems that practitioners genuinely care about. This grounding reduces ambiguity, enhances interpretability, and ensures verifiability of results. By contrast, series-centric generation often depends on abstract templates or rule-based slicing, which may produce trivial, poorly aligned, or even ambiguous tasks. Consequently, the problem-centric design not only raises the difficulty level but also improves the overall reliability, robustness, and meaningfulness of the benchmark.

Beyond the overall improvement in success rate under the problem-centric paradigm, several notable trends emerge. First, model performance is highly task-dependent: predictive tasks such as electricity forecasting exhibit relatively high success rates, while diagnostic tasks (e.g., anomaly detection with known anomaly rates) remain substantially more challenging, with larger variance in F1 scores. Second, although some models achieve high success rates, error-based metrics (e.g., MAPE, Absolute Error) reveal persistent numerical deviations, suggesting that LLMs capture coarse patterns but lack fine-grained precision. Finally, cross-domain comparison indicates that healthcare tasks are generally harder, while climate tasks demonstrate more stable performance. A detailed categorization of failure cases is further provided in Appendix E, offering insights into systematic error modes across tasks.

## 5 CONCLUSION

This work introduces the **TSAIA Benchmark**, a unified framework that combines series-centric and problem-centric perspectives to evaluate LLMs on time-series reasoning. Series-centric tasks capture canonical operations, while problem-centric tasks extend evaluation toward domain-grounded, multi-step analytical challenges.

Our findings show that existing models handle simple, template-based formulations relatively well but face substantial limitations in constraint adherence, compositional reasoning, and domain-specific signal integration. The problem-centric paradigm, despite introducing harder tasks, demonstrates higher success in task formulation and evaluation quality.

These results point to two key directions: (i) advancing LLMs toward stronger compositional reasoning and domain adaptation, and (ii) fostering problem-centric, dynamically extensible benchmarks to support progress in real-world time-series AI applications.

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

## A    THE USE OF LARGE LANGUAGE MODELS (LLMs)

In this work, we utilized LLMs as general-purpose assistants for time series analysis, particularly in the generation of executable code and the iterative refinement of model predictions. LLMs played a significant role in facilitating the generation of code for time series forecasting, anomaly detection, and other time series tasks, and were employed within a unified agent framework. This framework prompted the models to generate executable code, which was then executed, and refined based on feedback from the results of the model's performance. Our findings highlight both the strengths and limitations of current LLMs in performing time series analysis.

## B    DATASET STATISTICS

| Dataset | Number of Data Files | Avg Total Timestamps | Number of Variables |
|---|---|---|---|
| Climate Data | 624 | 526 | 2048 |
| Climate QA Data | 150 | 168 | 50 |
| Energy Data w/ geolocation | 22 | 8760 | 1-3 |
| Energy Data w/ Covariates | 66 | 872601 | 11 |
| Building Energy Usage Data | 398 | 5019 | 1 |
| Causal Data | 8 | 529 | 3–6 |
| Daily Stock Data | 6780 | 3785 | 7 |
| Hourly Stock Data | 5540 | 35 | 7 |
| Fresh Retail Data | 863 | 1440 | 17 |
| Stock Market Indices Data | 6 | 3388 | 4 |
| ECG Signal Data | 24 | 10804352 | 2 |
| ICU Clinic Data | 352 | 596 | 12 |

Table 5: Dataset Statistics of the constructed dataset. The exact number of time series are not calculated because it depends on randomly sampled sequence length when generating task instances.

Table 5 summarizes the dataset statistics for the raw time series datasets used in TSAIA. The climate data is obtained from ERA5 dataset [1]. The climate QA data is obtained from MTBench dataset [2]. Energy data with covariates is obtained from [3]. The ECG signal data and ICU Healthcare data are obtained from PhysioNet[4][5][6]. The building energy usage data is obtained from Kaggle[7]. The Fresh Retail Data is obtained from FreshRetailNet-50K dataset[8]. Notably, the daily stock data, hourly stock data, and energy data with geolocation were manually scraped and preprocessed. The energy data with geolocation was obtained from official energy grid operator websites[9][10][11], and the associated geolocation was inferred as the largest city within the operational zone delineated by each provider's published grid map[12][13][14]. Stock price data was scraped using the pyfinance[15] package, with data pulled up to date as of 2024-09-17. The stock market indices data are pulled from various sources on the web. The causal discovery dataset is synthetically generated to reflect controlled causal structures. The prompt used to obtain causal discovery dataset is shown in section G.

## C  COMPREHENSIVE RESULTS ON SERIES-CENTRIC TASKS

Tables 6–8 report the complete experimental results on the Series-Centric Tasks. These tasks cover three categories—predictive, diagnostic, and analytical—designed to evaluate LLMs across different aspects of time-series reasoning. For each subtask, we include the success rate along with task-specific error or accuracy metrics (e.g., MAPE, F1, Absolute Error), providing a fine-grained view of model performance. Results are compared across seven state-of-the-art models, with the best and second-best scores in each row highlighted in red and blue, respectively. Blank entries indicate results not yet available.

## D  DETAILS OF PROBLEM-CENTRIC MULTI-STEP TASK DESIGN

Table 9 summarizes the proposed Problem-Centric Multi-Step Task Design, where tasks are derived not merely from time series sequences but from real-world problems requiring domain-grounded reasoning. Unlike series-centric formulations that focus on direct prediction of future values, these tasks emphasize evidence gathering, operator-level reasoning, and structural decomposition to capture the complexity of decision-making in diverse domains. Specifically, retail tasks involve promotion-driven substitution ranking and stock-out–aware demand forecasting; clinical tasks target patient-level outcome prediction such as mortality and sepsis risk; and climate tasks cover both predictive (weather forecasting) and diagnostic (property damage and social impact) challenges. Each task is paired with domain-relevant metrics to ensure meaningful evaluation and fair comparison across models.

## E  ERROR ANALYSIS

Figure 5–9 present detailed error distributions across representative models. GPT-4o exhibits patterns broadly consistent with other systems: in predictive tasks, incorporating covariates or handling

---

[1] https://climatelearn.readthedocs.io/en/latest/user-guide/tasks_and_datasets.html#era5-dataset

[2] https://github.com/Graph-and-Geometric-Learning/MTBench

[3] https://github.com/tamu-engineering-research/Open-source-power-dataset

[4] https://physionet.org/content/nsrdb/1.0.0/

[5] https://physionet.org/content/ltdb/1.0.0/

[6] https://physionet.org/content/mimiciv/3.1/

[7] https://www.kaggle.com/competitions/energy-anomaly-detection/data

[8] https://huggingface.co/datasets/Dingdong-Inc/FreshRetailNet-50K

[9] https://www.nyiso.com/load-data

[10] https://www.ercot.com/gridinfo/load/load_hist

[11] https://www.misoenergy.org/markets-and-operations/real-time--market-data/market-reports

[12] https://www.nyiso.com/documents/20142/1397960/nyca_zonemaps.pdf

[13] https://www.ercot.com/news/mediakit/maps

[14] https://www.misostates.org/images/stories/meetings/Cost_Allocation_Principles_Committee/2021/Website_Presentations.pdf

[15] https://pypi.org/project/pyfinance/

| Task | Metric | GPT-4o | Qwen-Max | Claude-3.7 | DeepSeek | Gemini-2.5 | Codestral | Grok-4 |
|---|---|---|---|---|---|---|---|---|
| Electricity Prediction with Covariates — Max Load | Success Rate | 0.50 | 0.75 | | 1.00 | | 0.31 | |
| | MAPE (std) | 0.09 (0.12) | 0.07 (0.06) | | 0.10 (0.12) | | 0.03 (0.03) | |
| Electricity Prediction with Covariates — Min Load | Success Rate | 0.76 | 0.82 | | 0.88 | | 0.59 | |
| | MAPE (std) | 0.11 (0.11) | 0.09 (0.11) | | 0.12 (0.18) | | 0.09 (0.10) | |
| Electricity Prediction with Covariates — Load Ramp Rate | Success Rate | 0.46 | 0.80 | | 0.80 | | 0.47 | |
| | MAPE (std) | 0.18 (0.14) | 0.14 (0.12) | | 0.11 (0.08) | | 0.12 (0.08) | |
| Electricity Prediction with Covariates — Load Variability | Success Rate | 0.47 | 0.76 | | 0.76 | | 0.35 | |
| | MAPE (std) | 0.20 (0.31) | 0.13 (0.16) | | 0.19 (0.27) | | 0.04 (0.03) | |
| Electricity Prediction without Covariates — Max Load | Success Rate | 1.00 | 0.94 | | 0.94 | | 1.00 | |
| | MAPE (std) | 0.18 (0.16) | 0.10 (0.07) | | 0.15 (0.12) | | 0.12 (0.07) | |
| Electricity Prediction without Covariates — Min Load | Success Rate | 0.94 | 0.94 | | 0.88 | | 0.71 | |
| | MAPE (std) | 0.14 (0.08) | 0.14 (0.08) | | 0.13 (0.09) | | 0.14 (0.05) | |
| Electricity Prediction without Covariates — Load Ramp Rate | Success Rate | 0.76 | 1.00 | | 0.82 | | 0.88 | |
| | MAPE (std) | 0.24 (0.19) | 0.23 (0.22) | | 0.19 (0.20) | | 0.29 (0.30) | |
| Electricity Prediction without Covariates — Load Variability | Success Rate | 0.82 | 0.88 | | 0.76 | | 0.71 | |
| | MAPE (std) | 0.17 (0.12) | 0.13 (0.09) | | 0.19 (0.17) | | 0.13 (0.07) | |
| Electricity Prediction for Multiple Grids — Max Load | Success Rate | 0.76 | 0.88 | | 0.94 | | 0.12 | |
| | MAPE (std) | 0.21 (0.27) | 0.21 (0.24) | | 0.16 (0.21) | | 0.10 (0.03) | |
| Electricity Prediction for Multiple Grids — Min Load | Success Rate | 0.76 | 0.88 | | 0.94 | | 0.29 | |
| | MAPE (std) | 0.10 (0.12) | 0.18 (0.29) | | 0.08 (0.11) | | 0.23 (0.37) | |
| Electricity Prediction for Multiple Grids — Load Ramp Rate | Success Rate | 0.65 | 0.65 | | 0.94 | | 0.29 | |
| | MAPE (std) | 0.19 (0.24) | 0.18 (0.18) | | 0.21 (0.21) | | 0.10 (0.05) | |
| Electricity Prediction for Multiple Grids — Load Variability | Success Rate | 0.41 | 0.59 | | 0.59 | | 0.29 | |
| | MAPE (std) | 0.15 (0.13) | 0.18 (0.23) | | 0.18 (0.14) | | 0.19 (0.13) | |

Table 6: Full results on Series-Centric Tasks (Predictive Tasks). For rows with color highlights, red marks the best and blue marks the second best among the populated entries in that row. Blank cells indicate results not yet reported for that model.

multiple time series increases the likelihood of execution and constraint violation errors, reflecting the difficulty of enforcing operational limits under added complexity. Similarly, in diagnostic tasks, GPT-4o struggles when contextual reasoning is required—such as calibrating thresholds from reference samples—while tasks with explicit prior knowledge (e.g., causal discovery with known graphs) show comparatively higher success rates. In analytical tasks, GPT-4o's performance declines in market benchmarking and trading, where failures often stem from inadequate strategies or unfamiliarity with specialized financial metrics.

Across models, comparable limitations emerge. By contrast, Gemini-2.5 and Codestral suffer frequent execution errors across nearly all categories, underscoring difficulty with structured multi-step reasoning. Taken together, these results highlight that while model-specific differences exist, all systems—including GPT-4o—face substantial challenges as task complexity increases, especially when structured reasoning, contextual integration, or domain-specific financial acumen are required.

| Task | Metric | GPT-4o | Qwen-Max | Claude-3.7 | DeepSeek | Gemini-2.5 | Codestral | Grok-4 |
|---|---|---|---|---|---|---|---|---|
| Extreme Weather Detection with Reference Samples | Success Rate | 0.24 | 0.23 | | 0.23 | | 0.23 | |
| | F1 (std) | 0.91 (0.23) | 0.90 (0.24) | | 0.90 (0.24) | | 0.91 (0.23) | |
| ECG Signal Anomaly with Reference Samples | Success Rate | 0.51 | 0.17 | | 0.54 | | 0.59 | |
| | F1 (std) | 0.55 (0.35) | 0.70 (0.29) | | 0.54 (0.34) | | 0.58 (0.34) | |
| Causal Discovery with Quantitative Domain Knowledge | Success Rate | 0.94 | 0.92 | | 0.97 | | 0.94 | |
| | Accuracy (std) | 0.69 (0.09) | 0.77 (0.11) | | 0.71 (0.11) | | 0.72 (0.11) | |
| Causal Discovery with Qualitative Domain Knowledge | Success Rate | 0.85 | 0.70 | | 0.96 | | 0.93 | |
| | Accuracy (std) | 0.87 (0.17) | 0.79 (0.17) | | 0.89 (0.14) | | 0.88 (0.15) | |
| Extreme Weather Detection with Known Anomaly Rate (Across Sequences) | Success Rate | 0.87 | 0.31 | | 0.97 | | 0.23 | |
| | F1 (std) | 0.53 (0.25) | 0.62 (0.19) | | 0.72 (0.11) | | 0.42 (0.31) | |
| Energy Usage Anomaly with Known Anomaly Rate (Across Sequences) | Success Rate | 0.87 | 0.52 | | 1.00 | | 0.58 | |
| | F1 (std) | 0.08 (0.09) | 0.14 (0.20) | | 0.50 (0.19) | | 0.06 (0.06) | |

Table 7: Full results on Series-Centric Tasks (Diagnostic Tasks).

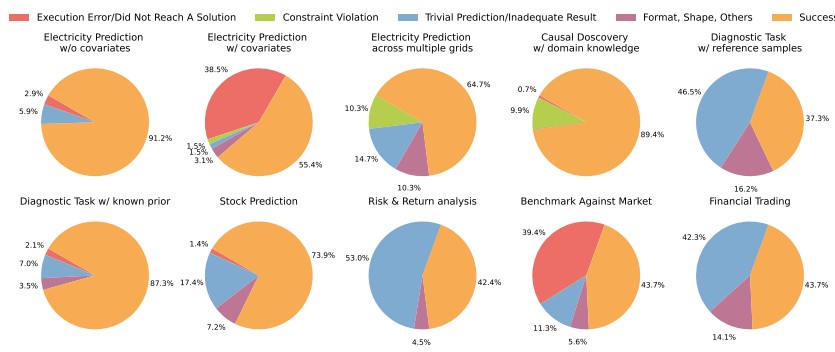

Figure 5: Case Study on GPT-4o Error Distribution across Tasks Grouped by Difficulty Level

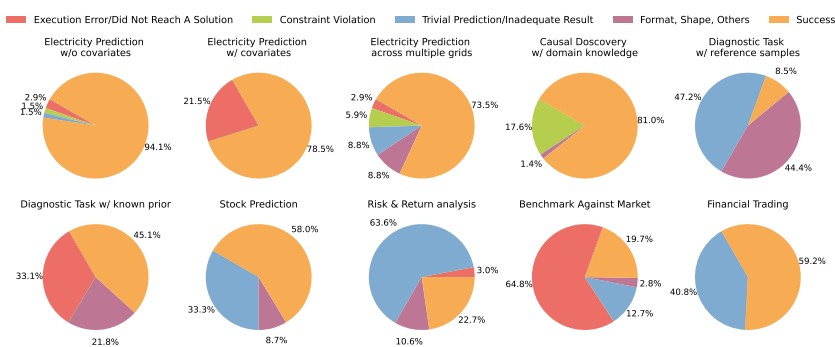

Figure 6: Case Study on Qwen Error Distribution across Tasks Grouped by Difficulty Level

| Task | Metric | GPT-4o | Qwen-Max | Claude-3.7 | DeepSeek | Gemini-2.5 | Codestral | Grok-4 |
|---|---|---|---|---|---|---|---|---|
| Future Stock Price Prediction | Success Rate | 0.96 | 1.00 | | 0.87 | | 0.39 | |
| | MAPE (std) | 0.06 (0.08) | 0.05 (0.07) | | 0.05 (0.07) | | 0.05 (0.05) | |
| Future Stock Volatility Prediction | Success Rate | 0.83 | 0.43 | | 0.57 | | 0.57 | |
| | MAPE (std) | 0.70 (0.28) | 0.83 (0.26) | | 0.90 (0.13) | | 0.61 (0.32) | |
| Future Stock Trend Prediction | Success Rate | 0.43 | 0.30 | | 0.43 | | 0.52 | |
| | Accuracy (std) | 0.90 (0.20) | 0.86 (0.23) | | 0.85 (0.23) | | 0.96 (0.14) | |
| Annualized Return Estimation | Success Rate | 0.45 | – | | 0.36 | | 0.18 | |
| | Absolute Error (std) | 0.02 (0.02) | – | | 0.02 (0.01) | | 0.03 (0.02) | |
| Annualized Volatility Estimation | Success Rate | 0.91 | 0.82 | | 1.00 | | 1.00 | |
| | Absolute Error (std) | 0.00 (0.00) | 0.00 (0.00) | | 0.00 (0.00) | | 0.00 (0.00) | |
| Maximum Drawdown Estimation | Success Rate | 0.18 | 0.09 | | 0.27 | | – | |
| | Absolute Error (std) | 0.00 (0.00) | 0.00 (0.00) | | 0.00 (0.00) | | – | |
| Calmar Ratio Estimation | Success Rate | 0.18 | 0.18 | | 0.27 | | 0.82 | |
| | Absolute Error (std) | 0.01 (0.01) | 0.01 (0.01) | | 0.02 (0.01) | | 0.01 (0.01) | |
| Sortino Ratio Estimation | Success Rate | 0.09 | 0.09 | | 0.18 | | 0.09 | |
| | Absolute Error (std) | 0.01 (0.00) | 0.01 (0.00) | | 0.00 (0.00) | | 0.00 (0.00) | |
| Sharpe Ratio Estimation | Success Rate | 0.73 | 0.18 | | 0.18 | | 0.73 | |
| | Absolute Error (std) | 0.00 (0.00) | 0.00 (0.00) | | 0.02 (0.02) | | 0.00 (0.00) | |
| Information Ratio (Benchmark vs. Market) | Success Rate | 0.44 | 0.20 | | 0.73 | | 0.01 | |
| | Absolute Error (std) | 0.00 (0.00) | 0.01 (0.01) | | 0.00 (0.00) | | 0.00 (0.00) | |
| Stock Trading Strategy Backtesting | Success Rate | 0.44 | 0.59 | | 0.61 | | 0.63 | |
| | Cumulative Return | 0.13 | 0.10 | | 0.09 | | 0.06 | |
| | Annualized Return | 2.43 | 4.56 | | 1.69 | | 3.87 | |
| | Maximum Drawdown | 0.05 | 0.05 | | 0.05 | | 0.02 | |

Table 8: Full results on Series-Centric Tasks (Analytical Tasks).

## F CODEACT SYSTEM PROMPT TEMPLATE

> You are a helpful assistant that gives helpful, detailed, and polite answers to the user's questions. The code written by assistant should be enclosed using <execute> tag, for example: <execute> print('Hello World!') </execute>. You should provide the solution in a single <execute> block instead of taking many turns. You'll receive feedback from your code execution. You should always import packages and define variables before starting to use them. You should stop <execute> and provide an answer when they have already obtained the answer from the execution result. Whenever possible, execute the code for the user using <execute> instead of providing it. Your response should be concise, but do express their thoughts. Always write the code in <execute> block to execute them. You should not ask for the user's input unless necessary. Solve the task on your own and leave no unanswered questions behind. You should do every thing by your self. You are not allowed to install any new packages or overwrite available variables provided to you in the question. Additionally, you are provided with the following variables available: {variable names} The above variables is already available in your interactive Python (Jupyter Notebook) environment, allowing you to directly use them without needing to re-declare them.

| Domain | Dataset | Problem | Level | Task Description | Metric |
|--------|---------|---------|-------|------------------|--------|
| Retail | FreshRetailNet-50K | Product Substitution Ranking | Operator | During promotion period of item A, rank the impact on all items in the same category. | NDCG, MAP |
| | | Future Demand Forecasting | Structural | First recover latent demand caused by stock-out, then forecast user demand over a future horizon. | WPE, WAPE |
| Healthcare | MIMIC-IV | Mortality Risk Assessment | Evidence | Given a recent window of ICU vitals/labs and related signals, estimate the patient's future mortality risk. | MAE |
| | | Sepsis Risk Prediction | Evidence | Given a recent window of ICU vitals/labs and related signals, estimate the patient's future sepsis risk. | ACC |
| Climate | MTBench | Weather Forecasting | Structural | Predict future weather values for multiple stations based on historical weather data during extreme events. | MAPE |
| | | Property Damage Prediction | Evidence | Predict property damage severity levels based on historical weather patterns and extreme weather characteristics. | ACC |
| | | Social Impact Prediction | Evidence | Predict social impact severity levels based on historical weather patterns and extreme weather characteristics. | ACC |

Table 9: Problem-Centric Multi-Step Task Design

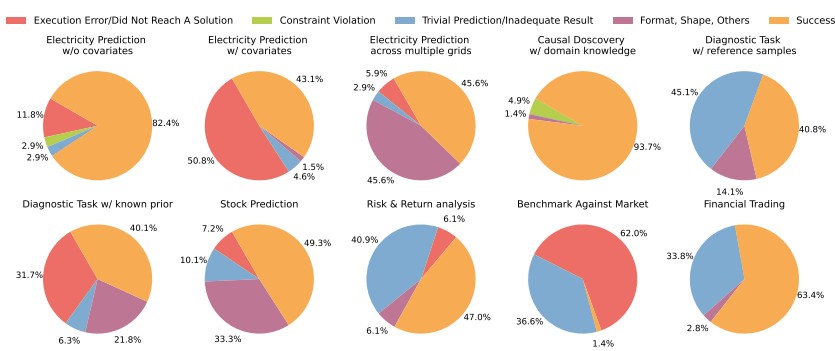

Figure 7: Case Study on Codestral Error Distribution across Tasks Grouped by Difficulty Level

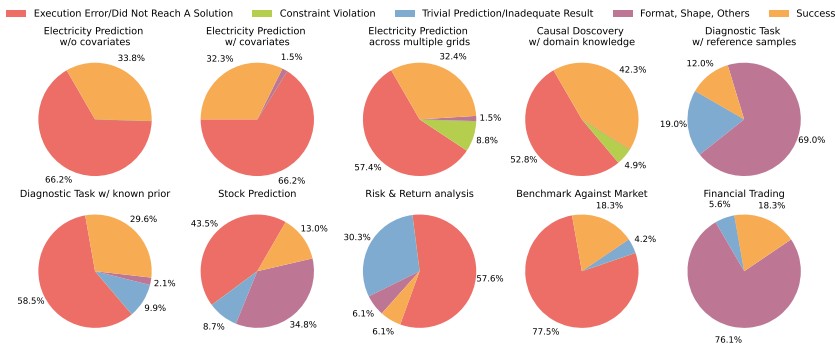

Figure 8: Case Study on Gemini Error Distribution across Tasks Grouped by Difficulty Level

# G    CASUAL DISCOVERY DATA GENERATION PROMPT

Now you are a Time series data scientist, please help me to write the code to generate some synthetic data in real world Time series domain, you should save the data into "*/data.csv":

Now suggesting you should construct a series data based on a relation matrix and the correlation ratio for different influence factor, you should notice the following points,for time step I want you to generate 500 time steps:

1. data correlation: the multi variable should be correlated, sample: which A first influence B, then B have influence on C or D, there should be some time delay, as the influence on other staff needs time.

2. data trend: there should be some trend in the data, like the data is increasing or decreasing.

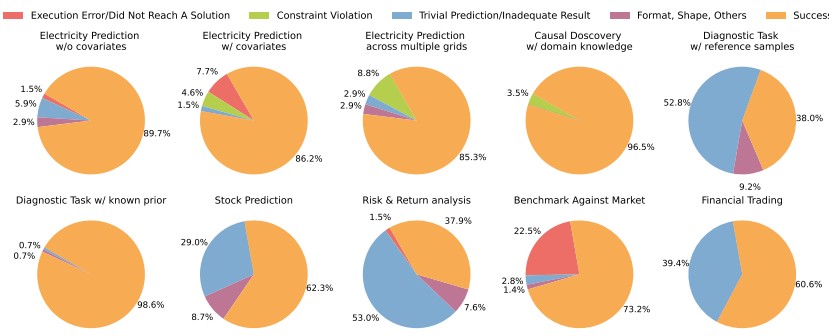

Figure 9: Case Study on Deepseek Error Distribution across Tasks Grouped by Difficulty Level

3. data: seasonality there should be some seasonality in the data, like the data is periodic.

4. data noise: the noise should be added to the data, as the real world data is not perfect.

5. data background: the data should have some real world background, you should first think about different real world data, and provide a description for the variable and time series data, then generate the data using the code. CoT Sample: Q: Approximate Relation Ratio: 0.5 Relation Matrix:

|   | A | B | C | D |
|---|---|---|---|---|
| A | 1 | 1 | 0 | 1 |
| B | 0 | 1 | 0 | 1 |
| C | 0 | 1 | 1 | 1 |
| D | 0 | 0 | 0 | 1 |

- A influences B and D, and itself.
- B influences D, and itself.
- C influences B and D, and itself.
- D influences only itself.

variable size: 4 A: Scenario: Sales Data of a Chain of Stores Over Time Let's assume we are generating synthetic data,the variable size for the data is 4. for the daily sales of multiple stores across a chain, the sales numbers are influenced by:

1. Advertising (A): The level of advertising spend directly impacts the sales of each store. After a delay, this starts influencing sales. 2. Sales (B): The sales numbers for each store are influenced by both the advertising and local seasonal events. 3. Economic Factors (C): Broader economic trends, like GDP growth or unemployment rates, also impact sales. These factors show a delayed and more subtle influence over time. 4. Customer Sentiment (D): Customer sentiment affects the sales of specific products in each store and is influenced by both advertising and broader economic factors.

Seasonality: Sales experience periodic seasonal trends, with peaks around the holidays and lower numbers during off-seasons.

Trend: There is a general increasing trend in sales as the chain expands.

Noise: Random noise is added to mimic real-world data fluctuations.

