# OpenReview forum: "When LLM Meets Time Series: A Real-World Benchmark for Explicit and Implicit Multi-Step Reasoning"
_ICLR.cc/2026/Conference — ICLR 2026 Conference Withdrawn Submission_

### Official Review · Reviewer_KhHg · 2025-10-29

**Soundness:** 2
**Presentation:** 3
**Contribution:** 2
**Rating:** 2
**Confidence:** 3

**Summary:**

This paper investigates the performance of Large Language Models (LLMs) on time series analysis tasks. The authors argue that current benchmarks used to evaluate LLMs in this domain have significant limitations, particularly in their ability to capture the diverse and complex nature of real-world time series problems. To address these limitations, the paper introduces a new benchmark named the Time Series Artificial Intelligence Assistant (TSAIA) Benchmark. The TSAIA Benchmark is designed to evaluate the capabilities of LLMs as time series artificial intelligence assistants. It is constructed using real-world datasets spanning multiple domains. The benchmark integrates two complementary suites of tasks: series-centric tasks and problem-centric tasks. The series-centric tasks focus on canonical time series analysis problems, providing continuity with existing benchmarks and enabling consistent baseline comparisons. In contrast, the problem-centric tasks are derived from real-world analytical scenarios across various domains, emphasizing applied reasoning and domain adaptation. The paper presents empirical results from seven LLMs evaluated on the TSAIA Benchmark. The findings indicate that no single model consistently outperforms the others across all tasks. Instead, the best-performing model varies depending on the specific task type and context.

**Strengths:**

- The paper examines the performance of LLMs on two complementary suites of time series analysis tasks: series-centric tasks and problem-centric tasks. This dual-task structure provides a comprehensive framework for assessing both fundamental and applied capabilities of LLMs in time series contexts.

- The series-centric tasks are designed to represent canonical time series problems that facilitate continuity with existing benchmarks and allow for meaningful baseline comparisons. These tasks encompass four key aspects of time series analysis: predictive, diagnostic, analytical, and decision-making

- The problem-centric tasks are constructed from real-world analytical problems across diverse domains. These tasks aim to test more complex reasoning and domain adaptation capabilities. They include: evidence integration, operator-based comparison, and structural multi-step reasoning.

- The paper presents the experimental results in Table 2 and Table 3, which summarize the performance of seven LLMs on both series-centric and problem-centric tasks. The results indicate that model performance varies across datasets and task types. There is no universally superior model, as the best-performing LLM depends on the specific dataset and nature of the task being evaluated.

- Overall, the paper is well-written, clearly organized, and accessible. It provides sufficient background and methodological context to understand the motivation, design, and contributions of the study. The discussion is coherent and allows readers to appreciate both the scope and limitations of the proposed benchmark.

**Weaknesses:**

- It would be beneficial for the paper to include a comparison with existing benchmarks in time series analysis. Although the paper highlights that current benchmarks have several limitations, the proposed benchmark is built on top of existing datasets and may therefore inherit some of these same limitations. A comparative analysis would help clarify to what extent the new benchmark addresses these issues. For example, it would be informative to examine the correlation between model performance on the proposed TSAIA Benchmark and on existing benchmarks. Such an analysis could strengthen the argument that TSAIA provides additional value or complementary insights beyond prior work.

- The paper could provide a more detailed evaluation of the quality and robustness of the proposed benchmark. Since the benchmark represents the paper’s main contribution, it is crucial to demonstrate its validity, reliability, and fairness. At present, it is difficult to fully assess these aspects based on the information provided. For instance, Line 215 mentions “verifiable reference output,” but the paper does not clearly explain how this verification process is implemented or validated. A more detailed description of the benchmark design, validation methods, and bias mitigation strategies would substantially improve the credibility of the contribution.

- The paper currently focuses primarily on analyzing LLM performance results, which provides a useful starting point for understanding model capabilities. However, the work could be made more impactful by also proposing or discussing approaches to improve LLM performance on time series analysis tasks. For example, including insights into model adaptation strategies, fine-tuning methods, or task-specific prompting approaches could transform the paper from a purely evaluative study into a more comprehensive contribution that advances methodology in this area.

- In Table 2, the color coding appears inconsistent. For instance, Grok-4 seems to achieve the best performance on the first task, but it is not highlighted in red as expected. Ensuring consistency in color representation across tables will improve clarity and prevent potential misinterpretations of the results.

- It would also be helpful to define more precisely the concept and scope of “reasoning” as used in the paper. Reasoning is a broad and often ambiguous term that can encompass several dimensions, such as logical inference, multi-step problem solving, or analytical reasoning. Providing a clear operational definition within the context of the benchmark would help readers interpret the results accurately and understand what types of reasoning the tasks are intended to evaluate.

**Questions:**

- It is important for the paper to include a thorough analysis of the quality of the proposed benchmark, as this is essential to ensure that the benchmark can have a meaningful and positive impact on the research community. Such an analysis should evaluate the robustness, fairness, and representativeness of the benchmark across different domains and datasets. Assessing these aspects would help determine whether the benchmark provides reliable and unbiased insights into the time series reasoning capabilities of LLMs.

- The paper should include a comparison between the proposed benchmark and existing benchmarks in time series analysis. This comparison would help clarify the specific contributions and advantages of the new benchmark. In particular, it is important to verify that the proposed benchmark does not inherit the same limitations identified in prior benchmarks, such as dataset bias, task redundancy, or a narrow scope of evaluation. Demonstrating that the benchmark effectively addresses these limitations would strengthen its validity and relevance to the community.

- It would be valuable for the paper to provide more details on how the task templates are constructed. Clarifying the process of template creation is important for ensuring transparency and reproducibility. In particular, the paper should explain how the task instructions are designed to prevent data leakage, especially the inadvertent inclusion of information about future data points. A clear description of the safeguards used to avoid such leakage would increase confidence in the integrity and reliability of the benchmark results.

---

### Official Review · Reviewer_Mrug · 2025-10-30

**Soundness:** 3
**Presentation:** 2
**Contribution:** 3
**Rating:** 6
**Confidence:** 3

**Summary:**

An agentic benchmark that evaluates large language models through practice-driven tasks integrated with a CodeAct based execution agent.

**Strengths:**

1. The tasks proposed to go beyond conventional time series analysis tasks such as forecasting or classification. They focus more on practical usage, such as constrained forecasting. These tasks are valuable for the community to think about.

2. The evaluation framework combines both traditional error metrics and a problem defined success rate, providing a clear picture of model performance.

Overall, thinking about more pipeline style tasks that mimic real world model deployment is an interesting direction. This work gives a concrete attempt to define several such tasks, which is encouraging.

**Weaknesses:**

1. The authors are encouraged to include a few actual examples (e.g., instruction, agent’s final output, evaluation criteria, and final score) to help readers better understand each task environment.
2. The current ablation only varies the LLM within a fixed CodeAct agent scaffold. However, changes in the scaffold itself could have a much larger impact on how the agent approaches and answers questions than simply switching the underlying LLM. Exploring scaffold variations or parameter settings could therefore provide deeper insights into where the performance differences truly come from.
3. Table 2 caption misses appendix reference
4. Figure 4 is could be hard to read. The author can consider adding a model/brand icon at the top of each bar for better visuals.

**Questions:**

Please see weakness

---

### Official Review · Reviewer_Ez61 · 2025-11-03

**Soundness:** 2
**Presentation:** 2
**Contribution:** 2
**Rating:** 2
**Confidence:** 5

**Summary:**

The paper proposes a benchmark to evaluate large language model based AI assistants on time series analysis. It combines series-centric tasks (e.g., forecasting, anomaly detection) with problem-centric tasks derived from real-world domains such as healthcare and climate science, emphasizing multi-step reasoning and evidence integration. Seven state-of-the-art LLMs are evaluated under unified, task-specific metrics. Results show strong performance on standard time-series tasks but significant weaknesses in reasoning, numerical precision, and constraint adherence, underscoring the need for more domain-grounded and extensible benchmarks.

**Strengths:**

The paper addresses the important and pressing need for real-world datasets to evaluate large language models, and foundation models' reasoning capabilities. I also really liked that the authors propose a dynamic benchmark, as well as the taxonomy of multi-step reasoning benchmarks (Section 2.2). I also like the formalization of the AI assistant in the form of a CodeAct agent.

**Weaknesses:**

1. **Positioning relative to prior work:** I would encourage the authors to compare their proposed benchmark with respect to many related benchmarks. For example both TimeSeriesGym and TimeSeriesExam are scalable, dynamic benchmarks and evaluate multi-step reasoning in LLMs and agents. Moreover, benchmarks used to evaluate foundation models such as MOMENT and TSPulse already include multiple different time series problems from different domains. The authors emphasize “dynamic extensibility,” however similar concepts have appeared in prior time-series and foundation model benchmarks.

2. **Improve the presentation of Section 2.2:** I really like Section 2.2, however some descriptions of the methods are unclear. I would encourage the authors to distill the high level take aways from the cited methods, so a reader who is unaware of the exact details of the cited studies, can still easily follow this section.

> Retail subtasks such as product substitution ranking and latent demand forecasting reveal difficulties in compositional reasoning.
3. How do you define compositional reasoning? I recommend looking at existing studies, such as [1] which formalize the notion of compositional reasoning, albeit in the context of forecasting.

4. **LLM-based AI assistants:** The notion of an AI assistant is unclear in the paper. The authors say that they are not evaluating agents, but do end up using an agentic scaffold (CodeAct). I would encourage the authors to clarify the class of methods (LLMs, tool-calling LLMs, multi-step agents) that their benchmark is designed for.

5. **Some important details are missing:**

> Trivial or degenerate responses—such as constant predictions, all-zero anomaly labels, or invalid code—are flagged as failures even when syntactically well-formed.

How do you automatically detect a trivial response? There's a lot of ways in which a response can be trivial, which also depends on the domain and task at hand.

> verification of constraint satisfaction and the use of required auxiliary information

How do you automatically verify the use of auxiliary information?

> Prediction is of correct shape and satisfies the specified operational constraint and the prediction is non-trivial (MAPE<1). A binary sequence with correct length is obtained and the prediction is non-trivial (F1-score>0)

I am not convinced that these correspond to "success". I also feel like the metric of success should be specific to each dataset.

> A scalar value is returned and the prediction is non-trivial (absolute error<0.05)

How was this threshold defined?

6. **Benchmark design:** The authors mention that their benchmark is extensible, however, it is unclear how their questions can be / are generated at scale. I also believe that a good benchmark should contain good baselines. The proposed benchmark only compares LLM-based models. Meanwhile, domain specific and agnostic statistical, deep learning, and time series foundation models may perform much better than the compared models. I believe that for this benchmark to be useful, it should also compare LLM-based models with these other approaches.

### References
1. Potosnak, Willa, et al. "Investigating compositional reasoning in time series foundation models." arXiv preprint arXiv:2502.06037 (2025).

**Questions:**

Please see the previous section for questions.

---

### Note · Authors · 2025-11-22

I have read and agree with the venue's withdrawal policy on behalf of myself and my co-authors.